

# An estimate of the area of occupancy and population size of *Brachycephalus tridactylus* (Anura: Brachycephalidae) to reassess its conservation status, with a proposal for conservation measures

Marcos R. Bornschein[1,2], Júnior Nadaline[3], Luiz Fernando Ribeiro[2], Giovanna Sandretti-Silva[1], Maria Fernanda Ferreira Rivas[1], Bruno de Morais Guerra[1] and Larissa Teixeira[1]

[1] Instituto de Biociências, Universidade Estadual Paulista, São Vicente, São Paulo, Brazil
[2] Mater Natura—Instituto de Estudos Ambientais, Curitiba, Paraná, Brazil
[3] Departamento de Zoologia, Universidade Federal do Paraná, Curitiba, Paraná, Brazil

Corresponding author
Marcos R. Bornschein,
marcos.bornschein@unesp.br,
bornschein.marcao@gmail.com

## ABSTRACT

**Background.** We are experiencing a global crisis in conservation, which has led to the prioritization of targets, such as nations, regions, and animal groups, which are necessary while resources are disputed. Brazil is a priority not only because of its megadiversity, high rates of endemism, and frequent descriptions of new species but also because of its high levels of deforestation. Among the species groups prioritized for conservation is the anurans (Amphibia: Anura), the population of which is severely declining. One group of anurans is the genus *Brachycephalus*, which includes 37 endemic species in the Brazilian Atlantic Rainforest. Some of these species have highly restricted distributions (<100 ha). Thirty new species have been described since 2000, and 55.3% of all species are threatened with extinction. *Brachycephalus tridactylus* was only recently described and remains restricted to its type locality. Because of its reduced geographical distribution (0.41 km$^2$), it has been proposed to be considered as Vulnerable. The objective of this study is to reevaluate the conservation status of *Brachycephalus tridactylus* and propose conservation measures.
**Methods.** We searched for new populations during 2016–2020, evaluated *in loco* impacts and potential impacts on the species' population, and performed an analysis of the density of this population and estimated its size. International Union for Conservation of Nature (IUCN) criteria were used to assess the conservation status of the species.
**Results.** We recorded the species in seven new localities (from 715–1,140 m above sea level) in the state of São Paulo up to 33 km from the type locality of the species (in state of Paraná). We estimated the area of occupancy as 148.44 km$^2$, densities as one calling male per 4.05 m$^2$ and 130.00 m$^2$, and a total population size of 4,429,722 adult individuals. Based on our finding, we proposed three lines of management: (1) formation of fire brigades, (2) management of residents' mules in the conservation unit and surrounding areas, and (3) management of degraded areas. We recommend changing the species' conservation status from Vulnerable to Endangered because of its fragmented distribution and decline in the area of occupancy and in the quality

of its habitat. Our results have expanded the species previous geographic distribution and delimited areas without previous records. Our estimates of population density and size are in accordance with those verified for congeners. The conservation of this species benefits the environments and other species that inhabit them, being, therefore, strategic for receiving conservation actions that will spread throughout the ecosystem.

## INTRODUCTION

Anura is a priority group for conservation assessments because of the high proportion of endangered species and rapid increase in descriptions of new taxa (*Tapley et al., 2018*). This priority is particularly urgent in countries with extensive deforestation, such as Brazil (*Tapley et al., 2018*). One of the Brazilian anuran endemic genera that has received increasing attention is *Brachycephalus* Fitzinger, 1826. It includes 37 endemic species of the Atlantic Rainforest (*Bornschein, Ribeiro & Pie, 2021*; *Frost, 2021*), 30 of which have been described since 2000. Equally striking is the percentage of species (55.3%) considered threatened with extinction (*Bornschein, Pie & Teixeira, 2019*). Moreover, 26.3% of the remaining species have been considered Data Deficient (*Bornschein, Pie & Teixeira, 2019*), which may be reassessed in the future as threatened with extinction. Finally, two species described after *Bornschein, Pie & Teixeira (2019)* have not yet been assessed for their conservation status (*Condez et al., 2021*; *Nunes et al., 2021*).

*Brachycephalus* includes small diurnal species (less than 2.5 cm in snout–vent length) that inhabit the leaf litter on forest floors and are usually heard rather than seen (*Pombal Jr & Gasparini, 2006*; *Condez et al., 2014*; *Ribeiro et al., 2015*; *Bornschein et al., 2016a*). Because the species depends on a cold, humid climate, they find suitable conditions at high altitudes, such as on mountaintops (*Pie et al., 2013*; *Bornschein et al., 2016a*). These high elevations are surrounded by lower altitudes with warmer climates, promoting speciation in the recent past (*Bornschein et al., 2016a*; *Firkowski et al., 2016*; *Pie et al., 2018a*; *Condez, Haddad & Zamudio, 2020*) and acting as sky islands (*sensu McCormack, Huang & Knowles, 2009*). This pattern explains the small geographic distributions of most species of *Brachycephalus* (*Bornschein et al., 2016a*), as well as other groups of reptiles, amphibians, and birds (*e.g.*, *Carnaval et al., 2014*; *Pulido-Santacruz et al., 2016*; *Oliveira et al., 2021*). The small distributions in conjunction with losses in areas of occupation and quality of habitats are the main cause of threats to the species (*Bornschein, Pie & Teixeira, 2019*).

One *Brachycephalus* species that was recently described and considered threatened with extinction (*Bornschein, Pie & Teixeira, 2019*) is *B. tridactylus*. It was described in 2012 based on seven individuals collected in 2007 and 2008 on the northern coast of Paraná at 900–930 m above sea level (a.s.l.; *Garey et al., 2012*). Subsequently, its type locality was corrected, and the altitude of occurrence changed to 880–910 m a.s.l. (*Bornschein et al., 2015a*). Based on new field data regarding the type locality, the recorded altitudinal range has been

increased to 805–910 m a.s.l. (*Bornschein et al., 2016a*); hence, the geographic distribution was estimated at 0.41 km$^2$ (*Bornschein et al., 2016a*). This small geographic distribution led to a proposal to classify the species as Vulnerable (*Bornschein, Pie & Teixeira, 2019*).

From 2016–2021, our research group conducted fieldwork on *Brachycephalus* around the type locality of *B. tridactylus*. In this article, we report the results of our research on the geographic distribution, population size, and impacts on habitat of this species. We also review and improve information known to support the conservation status of *B. tridactylus* and propose management strategies for its conservation.

## MATERIALS & METHODS

### Study region and field procedures

We delimited the maximum study region as a circle with a 60 km radius around the type locality of the species (according to *Bornschein et al. (2015a)*; Fig. 1). To improve the spatialization of the field effort, we delimited the approximate area of the Dense Ombrophilous Forest (Floresta Ombrófila Densa, *sensu Veloso, Rangel-Filho & Lima (1991)*) over 700 m a.s.l. (+/- 20 m) of this region, defined additional circles with radii of 10–50 km, and divided all the circles into four quadrants of 90° each (Fig. 1). We tried to sample most of the quadrants with forests from about 700 m a.s.l. The inaccessibility of mountainous and other high-altitude areas, however, limited the sampling. Although we conducted most of our research at sampling points that were accessible by secondary roads and trails, we often entered dense vegetation without trails. We conducted fieldwork from the end of 2016 to early 2020 in the spring and summer months (*i.e.,* from October to March), and also in September 2021.

At each locality of record, we took geographical coordinates using a GPS device (Garmin Etrex® 10 and Garmin GPSMAP® 60CSx; DATUM WGS84), identified and described the environmental impacts, and identified the type of vegetation according to the classification criteria for Brazilian vegetation (*Veloso, Rangel-Filho & Lima, 1991*). We were careful to classify the secondary vegetation that had not yet regenerated to the forest stage (see *Bornschein, 2015*). We recorded the altitude for each geographic coordinate using Google Earth Pro and rounded it to the nearest 5 m, according to *Bornschein et al. (2016a)*.

Specimens were anesthetized and euthanized using 2% lidocaine hydrochloride, fixed in 10% formalin, stored in a 70% ethyl alcohol solution, and deposited in the herpetological collections of the Museu de História Natural Capão da Imbuia (MHNCI), Curitiba, Paraná, Brazil. To ensure their identification, we compared these specimens with the original description of *B. tridactylus* (*Garey et al., 2012*) and with specimens we collected from its type locality and deposited in the MHNCI and the Célio F.B. Haddad collection (CFBH), Departamento de Zoologia, Universidade Estadual Paulista, Rio Claro, São Paulo, Brazil. We also compared advertisement call recordings of specimens with those from the type locality of *B. tridactylus* (*Bornschein et al., 2019a*). Collection permits for this study were granted by Instituto Chico Mendes de Conservação da Biodiversidade (ICMBio; #55918–1, #58088–2, #72845–1) and Secretaria do Meio Ambiente, Instituto Florestal, Coordenadoria de Tecnologia de Informação e Comunicação (COTEC; #693/2017 D63/2017 FN).

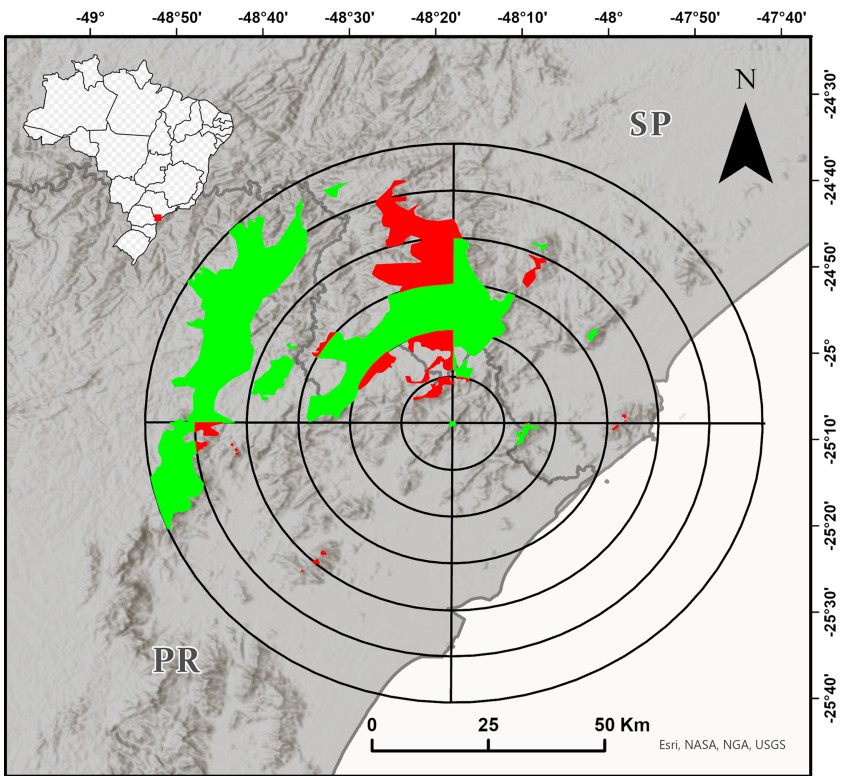

**Figure 1** **Target region for the search for new populations of *Brachycephalus tridactylus* divided into circles with radiuses of 10–60 km from the type locality of the species (*Bornschein et al., 2015a*) and subdivided every 90°.** The colored polygons represent the approximate distribution of the Dense Ombrophilous Forest above 700 m (+/- 20 m) above sea level (a.s.l.). The portions of the environment in each quadrant in which we carried out at least one field trip are shown in green, and the portions of the environment in each quadrant in which we did not carry out field trips are shown in red. Image sources: Esri, United States Geological Survey (USGS), National Geospatial-Intelligence Agency (NGA), National Aeronautics and Space Administration (NASA), and the GIS user community.

## Population size estimation

We estimated the number of calling males according to *Bornschein et al. (2016b)*, *Ribeiro et al. (2017)*, *Pie et al. (2018b)*, and *Bornschein, Teixeira & Ribeiro (2019)*. We delimited a polygonal area in which individuals were calling and used tapes to mark their positions. When no new materials were added after about 1 h, we counted all the tapes and measured the polygon, which resulted in an estimate of individuals per sampled area. Assuming a proportion of one female per male (*Bornschein, Teixeira & Ribeiro, 2019*), we extrapolated the density to the entire geographical distribution area of the species (*e.g.*, *Bornschein, Pie & Teixeira, 2019*). Since *Brachycephalus* are usually found in mountainous regions with difficult access, we did not randomize the sites for the census.

## Conservation

In the field, we observed the incidence of impacts or possible impacts that affected or could affect populations of *B. tridactylus* by making the habitat unavailable, reducing the quality of the habitat, and/or eliminating entire patches of habitat, as historical and recent losses of

forest coverage, fires, the suppression of vegetation beside roads, vegetation clearance below power lines, wood logging, edge effects, the construction of power plants and housing, the introduction of invasive alien plants, and extensive cattle ranching (*Bornschein et al., 2015b*; *Bornschein et al., 2016b*; *Garey & Provete, 2016*; *Bornschein, Pie & Teixeira, 2019*; *Bornschein, Teixeira & Ribeiro, 2019*; *Pie & Ribeiro, 2015*; *Ribeiro et al., 2015*; *Ribeiro et al., 2017*; *Pie et al., 2018b*; *Nunes et al., 2021*).

## Conservation status

We assessed the conservation status of the species according to *IUCN (2012)* and *IUCN Standards and Petitions Committee (2019)* criteria. However, we first determined the appropriate path for the assessment from the six pathways proposed by *Bornschein, Pie & Teixeira (2019)*. These paths allowed for possibility of assessment of the conservation status based on the number of records of the species, the knowledge of the altitudinal range of its occurrence, and the geographic distribution, regardless of whether it was confined to a mountain massif. Geographical distribution was determined in two ways: (1) by the space bounded by the lowest and highest registration altitude isolines and (2) by the space determined by connecting the extreme records using the minimum convex polygon method (MCP; *Mohr (1947)*). In both cases, inappropriate habitats, such as pasture lands, monoculture tree plantations, urban areas, croplands, and water surfaces, were removed from the distribution mapping, according to *Reinert, Bornschein & Firkowski (2007)*.

# RESULTS

## Habitat, geographic distribution, and altitudinal distribution

We found *Brachycephalus tridactylus* in seven new localities, all of which were situated in the state of São Paulo, southeastern Brazil (Table 1; Fig. 2). Vouchers (Table 1) showed a general orange color with green dots on the sides of the body (Fig. 3A), which aligned with the original description of coloration and with seven specimens we collected in the type locality (MHNCI 10294, 10729–30; CFBH 43887–90; Fig. 3B), and they presented an advertisement call with only isolated notes. Also, we did not record the species at 15 localities (Table S1).

We recorded the species at 715–1,140 m a.s.l. (Table 1) in Dense Montane in Dense Montane Ombrophilous Forest (Floresta Ombrófila Densa Montana), but we were unable to reach an altitude above 1,140 m a.s.l. We did not record the species in secondary vegetation not yet regenerated in forest, in transitional forest between Dense Montane between Dense Montane Ombrophilous Forest and Mixed Montane Ombrophilous Forest (Floresta Ombrófila Mista Montana), or in the Mixed Montane Ombrophilous Forest itself.

The seven new localities and the type locality of the species were enclosed in a minimum convex polygon (Fig. 4), from which we excluded areas at lower altitudes (40–714 m a.s.l.) or higher altitudes (1,141–1,300 m a.s.l.) than those of the recorded altitudinal range of the species, as well as areas where the original forest had been replaced or removed but was still regenerating and was not yet at the forest stage. This resulted in an area of occupancy

**Table 1** New localities of records of *Brachycephalus tridactylus* searched from 2016–2020. Abbreviation: MHNCI, Museu de História Natural Capão da Imbuia, Curitiba, Paraná, Brazil.

| Locality | Geographic coordinates[a] | Altitude (m) above sea level | | | Voucher |
|---|---|---|---|---|---|
| | | Searched for the species | With forests | Records of the species | |
| Bairro Rio Vermelho, Parque Estadual do Rio Turvo, municipality of Barra do Turvo, São Paulo | 24°59′25″S, 48°32′26″W | 660–835 | 770–820 | 770–790 | MHNCI 11643–45 |
| Estrada das Conchas, Parque Estadual do Rio Turvo, municipality of Barra do Turvo, São Paulo | 24°52′44″S, 48°19′42″W | 695–850 | 750–850 | 800–850 | MHNCI 11642 and recordings |
| Fazenda Fronteira, Parque Estadual do Rio Turvo, municipality of Barra do Turvo, São Paulo | 24°58′52″S, 48°16′54″W | 680–730 | 690–730 | 715–725 | MHNCI 11648–50, three uncataloged specimens, and recordings |
| Morro do Bisel, Serra do Guaraú, Parque Estadual do Rio Turvo, municipality of Cajati, São Paulo | 24°54′15″S, 48°12′55″W | 650–1,060 | 650–1,060 | 800–1,060 | MHNCI 11637 and recordings |
| Serra do Pinheiro, Parque Estadual do Rio Turvo, municipality of Cajati, São Paulo | 24°50′11″S, 48°16′11″W | 295–810 | 770–775 | 775 | MHNCI 11638–41 and recordings |
| Serra Pelada, Parque Estadual do Rio Turvo, municipality of Barra do Turvo, São Paulo | 24°58′50″S, 48°28′45″W | 740–1,140 | 1,130–1,140 | 1,130–1,140 | MHNCI 11571, 11646–47, one uncataloged specimen, and recordings |
| Torre Embratel, Parque Estadual do Rio Turvo, municipality of Cajati, São Paulo | 24°52′46″S, 48°15′27″W | 950–1,000 | 950–1,000 | 960–990 | MHNCI 10848, 10852, 11630–36, 11 uncataloged specimens, and recordings |

**Notes.**
[a]DATUM WGS84.

of 148.44 km$^2$ (14,843.6 ha; Fig. 4), 78.2% (116.0 km$^2$) of which was in São Paulo state, and the remaining area (32.4 km$^2$) in Paraná state.

## Density and population size

At Torre Embratel (see Table 1), we counted calling males within a polygon of 32.4 m$^2$ (24°52′46″S, 48°15′30″W) for 3 h (9:50–12:50 h) on January 27, 2018 and in a polygon of 1,300 m$^2$ (24°52′45″S, 48°15′27″W) for almost 3 h (9:50–12:40 h) on September 7, 2021. There were 8 and 10 calling males per counting—one per 4.05 m$^2$ and 130.00 m$^2$, respectively (a mean of one male per 67.02 m$^2$). Assuming a proportion of one female per male, the density of one adult individual per 33.51 m$^2$ implied a population of 4,429,722 adult individuals.

## Impacts on populations

The largest part of the area of occupancy of *B. tridactylus* was within the Parque Estadual do Rio Turvo (State Park Turvo River), in the municipalities of Barra do Turvo and Cajati, São Paulo (112.8 km$^2$; 75.7% of total area of occupancy). Approximately 0.24% (3.3 km$^2$) was within the Reserva Natural Salto Morato (Natural Reserve Salto Morato)—a private park in the municipality of Guaraqueçaba, Paraná. The areas with adequate conditions that could have been included in the area of occupancy of the species, but were not due to

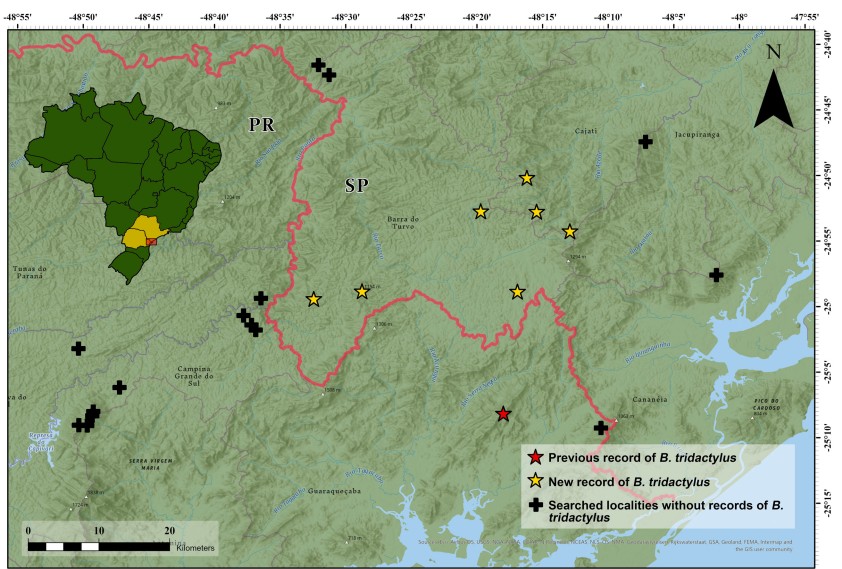

**Figure 2** **Geographical distribution of *Brachycephalus tridactylus* (stars) in southeastern and southern Brazil.** Type locality (red star) according to *Bornschein et al., 2015a* and *Bornschein et al., 2015b*. Abbreviations: SP, São Paulo; PR, Paraná. Image sources: Esri, Airbus Defence and Space (Airbus DS), United States Geological Survey (USGS), National Geospatial-Intelligence Agency (NGA), National Aeronautics and Space Administration (NASA), Consultative Group on International Agricultural Research (CGIAR), N Robinson, National Center for Ecological Analysis and Synthesis (NCEAS), National Library Service (NLS), Ordnance Survey (OS), New Masters Academy (NMA), Geodatatyrelsen, Rijkswaterstaat, GSA Technology Deployment Maps, Geoland, Federal Emergency Mgmt Agency (FEMA), Intermap, and the GIS user community.

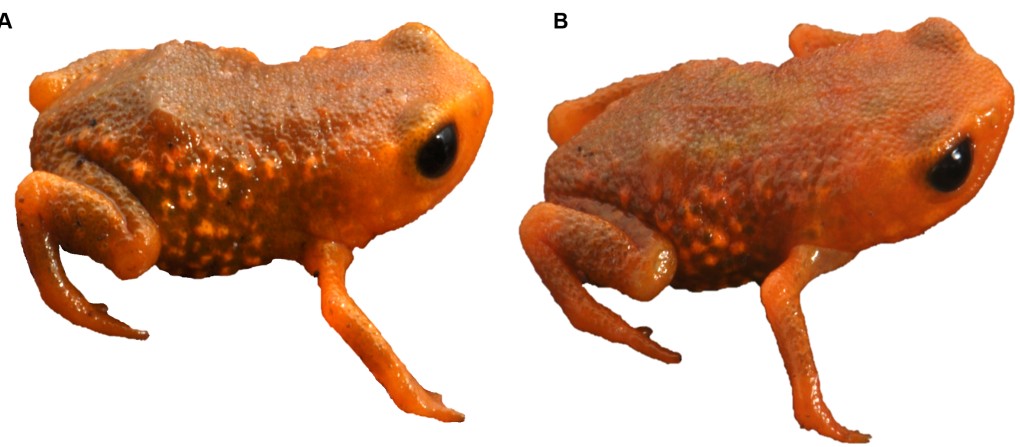

**Figure 3** *Brachycephalus tridactylus.* **from Morro do Bisel (A), Cajati, São Paulo (MHNCI 11637) compared with one individual from its type locality (Reserva Natural Salto Morato (B), Guaraqueçaba, Paraná (CFBH 43887)).** Photographs: Luiz F. Ribeiro.

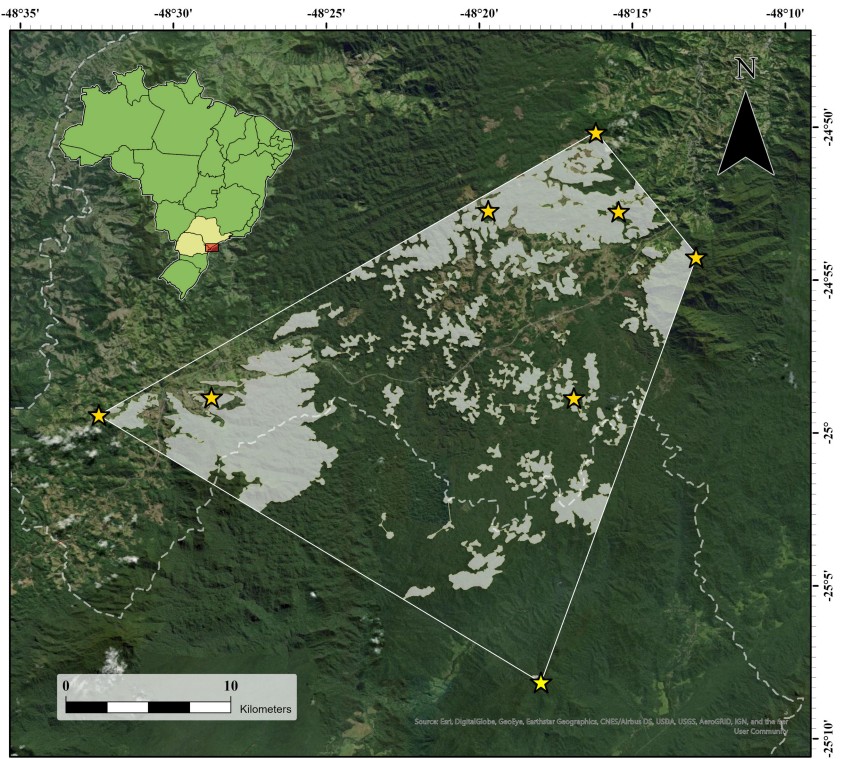

**Figure 4** **Area of occupancy of *Brachycephalus tridactylus* (light gray polygons) inside the minimum convex polygon connecting (solid white line) the extreme records in southeastern and southern Brazil.** The white dashed line represents the state border between São Paulo (northeastern portion) and Paraná. Image sources: Esri, DigitalGlobe, GeoEye, Earthstar Geographics, Centre National d'Etudes Spatiales (CNES)/ Airbus Defence and Space (Airbus DS), United States Department of Agriculture (USDA), United States Geological Survey (USGS), AeroGRID, Institut Geographique National (IGN), and the GIS user community.

deforestation, were in the state of São Paulo within the limits of the Parque Estadual do Rio Turvo. In this conservation unit, there was a historical reduction in the area of occupancy of the species due to swiddens, conversions to pasture land, constant losses due to the spread of forest fires started for the revitalization of pasture land, and logging (Fig. 5). Also, a recent impact has been a reduction in the quality of the habitat. Forest areas on the edge of deforested areas suffer from the edge effect, which locally reduces the occurrence of *B. tridactylus* by a few meters from the edge toward the forest interior.

We highlighted the intense impacts of *palmiteiros* (palm heart [*Euterpe edulis* Mart.] harvesters; Fig. 6) who (i) cut almost all adult individuals of this species of palm, (ii) change the structure of the lower strata of vegetation to cut the palm hearts, and (iii) open relatively wide trails to allow the passage of mules and their loads of palm hearts, which are transported in bundles on the sides of the animals. There is no area in the eastern Atlantic Forest where palm hearts are not targeted by *palmiteiros*, even in conservation units. In all the areas that we accessed on the Parque Estadual do Rio Turvo, we saw cut palm hearts, even away from roads. To access up to 1,080 m of altitude from Morro do

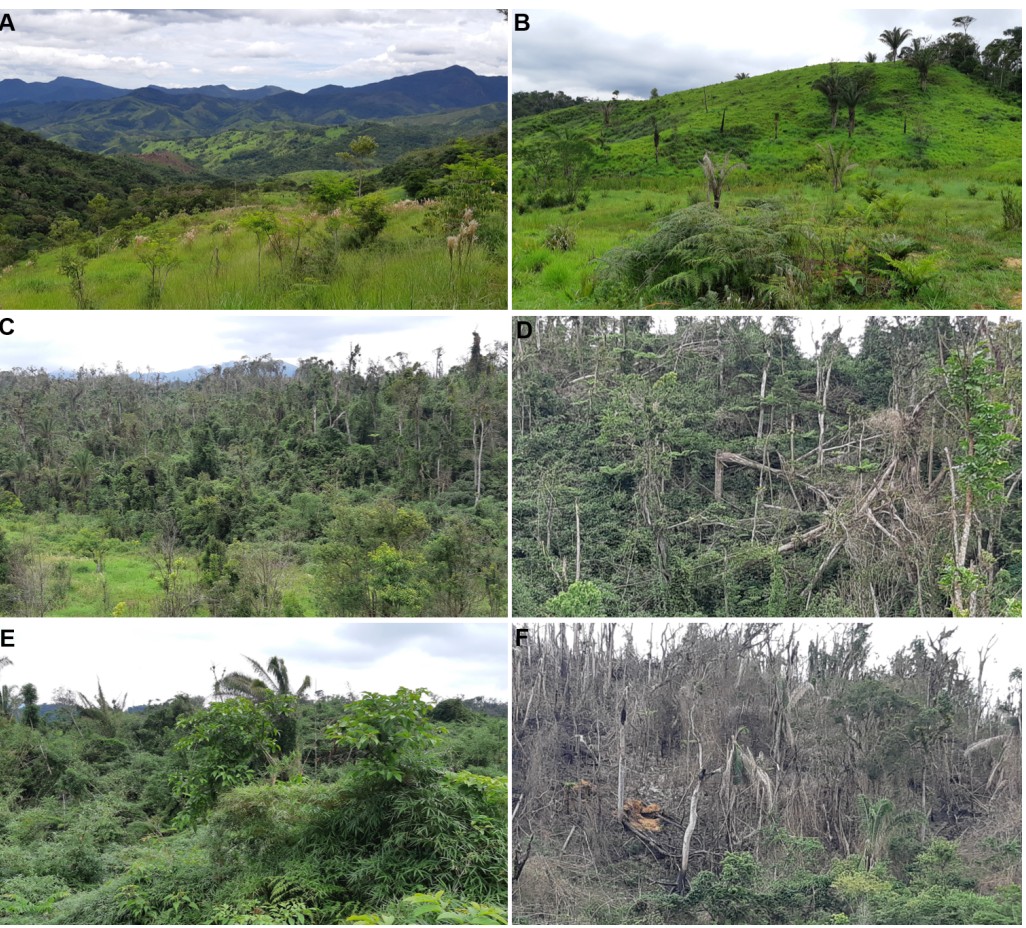

**Figure 5** **Habitat loss of *Brachycephalus tridactylus* in the Parque Estadual do Rio Turvo, São Paulo.** (A) A general view of the landscape around the BR 116 highway (crossing the second plane from left to right), highlighting deforested areas that are maintained by regular fires. (B) Close-up view of pastures maintained by fire. (C) A forest greatly altered by fire, wood logging, and edge effects. (D) A forest deeply altered by wind. (E) Secondary vegetation with ecological succession impaired by dense bamboo cover. (F) Arboreal vegetation destroyed by fire. Photographs: Marcos R. Bornschein (2019).

Bisel, for example, we walked 2.1 km into the forest without trails, where we saw cut palm hearts up to 950 m a.s.l. We also observed frequently used *palmiteiro* trails and campsites up to 900 m a.s.l.

## Conservation status

Based on the IUCN criteria for classifying the conservation status of species, we propose that *B. tridactylus* should be classified as Endangered. The criteria are as follows: geographic range (B) in the form of area of occupancy (B2) are severely fragmented (a; Fig. 4) with continuous decline observed, inferred, or projected (b) in the area of occupancy (ii), and area, extent and/or quality of habitat (iii) [EN B2ab(ii,iii)].

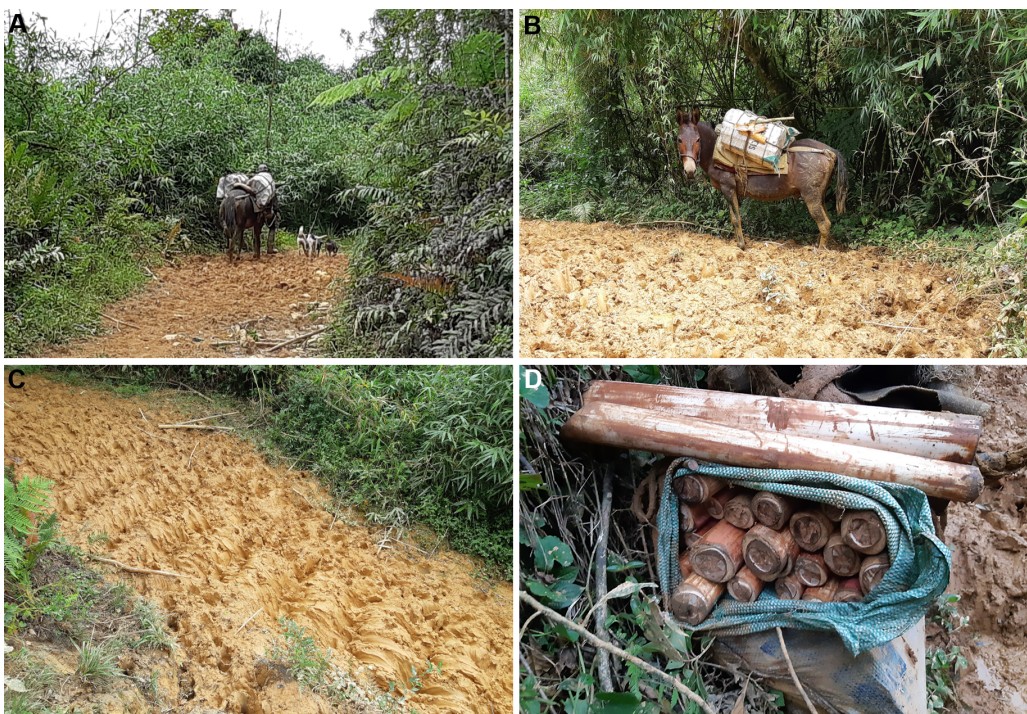

**Figure 6** **Intense illegal palm heart cutting (of *Euterpe edulis*) by *palmiteiros* along an old rural road in a region with records of *Brachycephalus tridactylus* in the Parque Estadual do Rio Turvo, São Paulo.** (A) *Palmiteiro* with one of two mules bearing a load of about 60 palm hearts. (B) The *palmiteiro*'s second mule also carrying a load of about 60 palm hearts. (C) Characteristic mud formed by the frequent passage of mules, making depressions where their feet step between elevations that correspond to the length of their strides. (D) About 55 palm hearts left on the edge of the rural road because of the rupture of the ropes supporting the load on a mule. Photographs: Marcos R. Bornschein (2019).

## DISCUSSION

### Geographic distribution

The systematic search resulted in several new records that extended the geographical distribution of *B. tridactylus* 33 km to the north. The presence of *B. tridactylus* in the south of São Paulo was in an adequate climate continuous with the mountains and plateaus in Paraná (see Fig. 3e in *Pie et al., 2013*).

We conducted extensive sampling in the vast study region—a circle with a 60 km radius around the type locality of the species (Fig. 1). Also, the species was unrecorded in another 15 localities (Table S1), suggesting that its current estimated geographic distribution is realistic. Large distributional extensions are not expected to be detected in future research. Although there is an environmental continuum in the southwest of the study region, where the species could be recorded (Fig. 1), the area is occupied by *B. brunneus*, which has no indicated sympatry with *B. tridactylus* (*Bornschein, Pie & Teixeira, 2019*). New localities of records, however, should be made within the study region, perhaps in the Serra da Virgem Maria (25°06′38″S, 48°31′42″W), on the border of São Paulo and Paraná states, at 1,487 m a.s.l. The area has never been explored by scientists. The difficulty in accessing mountains

limits the search for new populations of *Brachycephalus*, since it requires bold explorations similar to the exploration we conducted to reach Morro do Bisel. Our team conquered and named this mountain after two years of searching for the best access point, which required developing 2.1 km of trails.

The occurrence of *Brachycephalus tridactylus* overlaps that of *B. sulfuratus* at least in Bairro Rio Vermelho, Morro do Bisel, Serra do Pinheiro, and Torre Embratel. These species have been registered less than 20 cm from each other. There are a few cases of sympatry between *Brachycephalus* species (*Bornschein et al., 2016a*; *Bornschein et al., 2021*), most of which include *B. sulfuratus* (*Bornschein et al., 2016a*).

## Density and population size

The density of calling males per m$^2$ estimated for *B. tridactylus* resembles that of *B. curupira* in areas with lower abundances of individuals (*Ribeiro et al., 2017*); is lower than that of *B. albolineatus* at higher altitudes (one calling male for every 3–4 m$^2$, *Bornschein et al. (2016b)*), *B. mirissimus* (one calling male for every 14.5 m$^2$; *Pie et al. (2018b)*), and *B. fuscolineatus* (one calling male for every 4 m$^2$, *Bornschein, Teixeira & Ribeiro (2019)*); and is higher than that of *B. albolineatus* at lower altitudes (one calling male for every 100 m$^2$; *Bornschein et al. (2016b)*). The population size of *B. tridactylus* is greater than that of *B. albolineatus*; lower than that of *B. didactylus*, *B. sulfuratus*, and *B. pitanga*, and well above that of, *B. fuscolineatus* (*Bornschein, Pie & Teixeira, 2019*).

## Conservation

*Brachycephalus tridactylus* occurs in forests, but we did not record this species at forest edges in deforested areas. The same was verified for *B. fuscolineatus* (*Bornschein, Teixeira & Ribeiro, 2019*), possibly due to microclimate changes, such as the reduced humidity averages and increased temperature averages caused by the disruption of forest structure at the edges (*Magnago et al., 2015*). We also did not record *B. tridactylus* in regenerating tree formations that had not yet reached the forest stage. This finding agreed with most occurrences of species of the genus; only five species have been recorded in secondary vegetation not yet regenerated to a forest (*Bornschein et al., 2019b*).

The revision of the conservation status of *B. tridactylus* occurred through pathway #5 of *Bornschein, Pie & Teixeira (2019)*. This scrutiny changed the previous classification of the species as Vulnerable (*Bornschein, Pie & Teixeira, 2019*) to Endangered, a higher threat category. The cause of threat for the species was previously limited to the reduced geographical distribution, but reduced habitat and reduced quality of habitat have also now been incorporated as threats. Both factors have also affected other species of the genus, such as *B. mariaeterezae* and *B. quiririensis* (*Bornschein, Pie & Teixeira, 2019*). Several areas that could have contributed to the area of occupancy of *B. tridactylus* have been entirely lost or overexploited, even in conservation units (Parque Estadual do Rio Turvo), confirming that the simple creation of conservation units is not sufficient to ensure the effective protection of biodiversity but also requires adequate management. To reverse the situation, the lands included in conservation units of integral protection must fully pass to the state through expropriation and compensation, allowing them to be managed and regenerated naturally and/or through management actions.

Land tenure regularization depends on the political will to prioritize the allocation of resources, which is difficult to achieve (*Schiavetti & Santos, 2012*). The historical economic difficulties in Brazil and the constant trend toward the privatization of conservation units have postponed expropriations. Because of the recent economic crisis caused by the Covid-19 pandemic, expropriations may even be absent from the government's agenda for years to come. Brazil is among the nations that have eased conservation actions and laws in the wake of the pandemic (*Vale et al., 2021*). It is likely that the return to the pre-pandemic state investment in the environment and respect for environmental guidelines by the population will take decades. Since the state has reduced its involvement, environmental violations have increased in the rural population; for example, we heard openly about the daily illegal exploitation of palm hearts by *palmiteiros* on an old secondary road (see *Díaz et al., 2019*).

The areas around the preserved forests (mainly the secondary vegetation) are dominated by the African grass *Megathyrsus maximus* (Jacq.) B.K.Simon & S.W.L.Jacobs—a species characterized by dense biomass of great height (up to 3 m)—which the communities manage only by fire. Many fires are also caused by cigarettes thrown from vehicles and by uncontrolled burning in fields. These fires gradually enter the forest, first burning the edge, but thereafter killing trees that later become dry biomass for subsequent fires, which can advance further into the forest, and so on. The Atlantic Rainforest is humid, but the mountains are more prone to humidity fluctuations because of the stronger winds, greater intensity of solar radiation, and brighter illumination when clouds are absent (*Roderjan, 1994*; *Rizzini, 1997*). The humidity in montane forests can vary widely daily, and periods of low humidity can persist for days, causing water stress in plants (*e.g.*, *Roderjan, 1994*; *Rapp & Silman, 2012*). The synergistic effects of drier years with an increased frequency of fires can favor the movement of forest fires toward pristine mountainous areas, where slopes are accentuated and favor the humidity loss.

Degraded areas dominated by bamboo (Fig. 5E) tend not to advance in natural recovery but regress in their ecological succession when they are dominated by these fast-growing native plants. Because its stems are emitted successively on the branches and tops of native trees, the accumulated weight tends to cause physical damage by breaking the tree branches and trunks, arresting the succession (*Griscom & Ashton, 2006*). Areas dominated by bamboo are common in the regions that *B. tridactylus* inhabits. Bamboo may be the habitat of specialist species, such as rats and birds (*e.g.*, *Cockle & Areta, 2013*). Of the various bamboo birds that occur in the regions that *B. tridactylus* inhabits, such as *Scytalopus speluncae* (taxonomy according *Maurício et al. (2010)*), *Anabazenops fuscus*, and *Biatas nigropectus* (*Bornschein, Ribeiro & Pie, 2021*, pers. obs.), the latter two occur particularly in conjunction with *Guadua angustifolia* Kunth, which is common in the region of occurrence of *B. tridactylus*. *Biatas nigropectus* has been identified by IUCN as threatened with extinction (*BirdLife International, 2018*) and described as Vulnerable, but it is not threatened according to the Brazilian list ("Portaria MMA No 444", December 17, 2014); thus, there are no requirements for the management of bamboo because of its potential occupation by endangered species.

We suggest intensifying private sector involvement and directing resources from environmental fines as a way to solve environmental problems in the Parque Estadual do Rio Turvo. The degraded area in this conservation unit over 715 m a.s.l., which meets the altitudinal criterion regarding the area of occupancy of *B. tridactylus*, covers 17.2 km$^2$. This is the extent of area that could be reclaimed to support the improved geographic distribution of the species.

## Conservation measures proposed

We recommend conservation management on three fronts: (1) the formation of fire brigades, (2) the management of residents' mules in conservation units and surrounding areas, and (3) the management of degraded areas. Permanent firefighting teams would prevent fires by introducing fire-breaks. Secondary vegetation areas are burned every year or at intervals of two or three years in winter, especially in July and August, and in extremely hot summer months (*e.g.*, February). Fires are abundant near communities and along several kilometers of the BR 116 highway (Figs. 5A and 5B).

We suggest placing microchips and sealed radio collars on the residents' mules in the Parque Estadual do Rio Turvo and surrounding areas to control the activities of *palmiteiros* and reduce the impact of animals on vegetation through monitoring. The data gathered by the radio collars should be downloaded regularly and analyzed to check the animals' movements. This proposal must be supported by public policy that would allow the monitoring of animals. Moreover, it must be conducted in partnership with the local police, who would be able to act on proof that mules were entering the state park.

Regarding the management of degraded areas, we propose an intervention on two fronts: sowing degraded areas and reducing bamboo biomass to facilitate subsequent sowing. We propose the sowing of seeds directly by men in the field or by air in two stages: (1) an initial intervention and (2) when the sown vegetation has grown to 1.5 m to 2 m in height around two to four years after sowing. For the first intervention, we propose sowing seeds of the *Leandra australis* (Cham.) Cogn. and *Piper aduncum* L. shrubs in combination with three to six trees species from *Alchornea glandulosa* Poepp. & Endl., *A. triplinervia* (Spreng.) Müll.Arg., *Citharexylum myrianthum* Cham., *Miconia cinerascens* Miq., *M. cinnamomifolia* (DC.) Naudin, *M. formosa* Cogn., *Myrsine coriacea* (Sw.) R.Br. ex Roem. & Schult., *Psidium guajava* L., *Schizolobium parahyba* (Vell.) Blake, *Senna multijuga* (Rich.) H.S.Irwin & Barneby, *Tibouchina pulchra* Cogn., *Trema micrantha* (L.) Blume, and *Vochysia bifalcata* Warm. The shrubs have excellent coverage and would help to reduce the dominance of the African grass *M. maximus*. For the second intervention, we propose dispersing seeds from trees grown in the first intervention, particularly *A. glandulosa* and *A. triplinervia*. We also suggest incorporating seeds from the trees *Didymopanax morototoni* (Aubl.) Decne. & Planch., *Hyeronima alchorneoides* Allemão, and *Piptadenia gonoacantha* (Mart.) J.F.Macbr., as well as palm seeds from *Syagrus romanzoffiana* (Cham.) Glassman. Additional sowing should be evaluated *in loco* according to vegetation development. In the event of fires in the intervention areas, the first stage would be repeated.

We suggest prioritizing recovery areas that are dominated by bamboo on the margins of forests in a good stage of conservation and/or close to places where there are regular fires.

It would be necessary to construct a grid of trails (perhaps 5 m from each other) using scythes to cut the bamboo at the base. This process should be repeated every three months to cut the new shoots while they are still soft. As the bamboo biomass starts to decompose and access between the parallel trails becomes easier, the bases of the remaining bamboo should be cut, and sowing should then begin as described above.

## CONCLUSIONS

*Brachycephalus tridactylus* is currently threatened with extinction especially due to its restricted geographic distribution caused by habitat reduction and the loss of habitat quality. Despite this, this species has a high density and a high population size, like the congeners for which these data are available. The conservation of this small toadlet requires local measures, such as the restoration of degraded areas, as well as comprehensive actions, such as cultural changes regarding the use of fire as an agricultural practice. Furthermore, the chain of action proposed for conservation should include both local communities and the state, including the development of public policy. These actions and their effects could go beyond the area of occupancy of the species, benefiting many other species; therefore, we propose the concept of a fountain species that includes small species, the conservation of which would affect many flora and fauna, like a source of water that is local but spreads widely.

## ACKNOWLEDGEMENTS

Tamiris Pereira-Lima and Andre Leite helped with the fieldwork. The managers of Núcleo Capelinha in the Parque Estadual do Rio Turvo provided accommodation and logistical support during the fieldwork in Serra do Pinheiro. Kelsey Neam and two anonymous reviewer made valuable comments that improved the quality of the text.

### Funding
The field work was funded by the National Geographic Society (grant EC-50722R–18) and Fundação Grupo Boticário de Proteção à Natureza (1149_20191) through a project conducted by Mater Natura –Instituto de Estudos Ambientais. Júnior Nadaline received a grant from CAPES/Programa de Excelência Acadêmica –PROEX (process 88887.314020/2019-00); Giovanna Sandretti-Silva received a grant from PIBIC (process 159749/2020–4); Larissa Teixeira received grants from PIBIC (process 43278), FAPESP (process 17/21611–9), and CAPES/REITORIA (process 88882.441704/2019-01). There was no additional external funding received for this study. The funders had no role in study design, data collection and analysis, decision to publish, or preparation of the manuscript.

### Grant Disclosures
The following grant information was disclosed by the authors:
The National Geographic Society: EC-50722R–18.

Fundação Grupo Boticário de Proteção à Natureza (1149_20191) through a project conducted by Mater Natura –Instituto de Estudos Ambientais.
CAPES/Programa de Excelência Acadêmica –PROEX: process 88887.314020/2019-00.
PIBIC: process 159749/2020–4 and process 43278.
FAPESP: process 17/21611–9.
CAPES/REITORIA: process 88882.441704/2019-01.

## Competing Interests

The authors declare there are no competing interests.

## Author Contributions

- Marcos R. Bornschein conceived and designed the experiments, performed the experiments, analyzed the data, prepared figures and/or tables, authored or reviewed drafts of the paper, and approved the final draft.
- Júnior Nadaline, Luiz Fernando Ribeiro, Maria Fernanda Ferreira Rivas, Bruno de Morais Guerra and Larissa Teixeira performed the experiments, analyzed the data, prepared figures and/or tables, and approved the final draft.
- Giovanna Sandretti-Silva performed the experiments, analyzed the data, prepared figures and/or tables, authored or reviewed drafts of the paper, and approved the final draft.

## Animal Ethics

The following information was supplied relating to ethical approvals (i.e., approving body and any reference numbers):

Our study did not involve animal testing or other laboratory animal studies. So, there is no need to obtain an animal care license in Brazil.

## Field Study Permissions

The following information was supplied relating to field study approvals (i.e., approving body and any reference numbers):

Instituto Chico Mendes de Conservação da Biodiversidade (ICMBio) and Secretaria do Meio Ambiente, Instituto Florestal, Coordenadoria de Tecnologia de Informação e Comunicação (COTEC).

## Data Availability

The raw data is in the Supplemental File.

## Supplemental Information

Supplemental information for this article can be found online at http://dx.doi.org/10.7717/peerj.12687#supplemental-information.

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
