# Peer review of "An estimate of the area of occupancy and population size of Brachycephalus tridactylus (Anura: Brachycephalidae) to reassess its conservation status, with a proposal for conservation measures"

_PeerJ, doi:10.7717/peerj.12687_

## Round 0.1 · original submission · Major Revisions

Your paper was carefully reviewed by three reviewers. All of them pointed out problems with your population estimate. Please consider all comments in your next revision. In addition, check the language to make sure the manuscript flows well and is free of inconsistencies.

Reviewer 1 ·

Basic reporting

Some methods need to be clarified. In particular, I could not follow the sampling scheme that occurred at each site (Lines 94-95).

Experimental design

As mentioned above, I did not follow all of the methods. More detail would need to be provided for them to be replicated.

Validity of the findings

My biggest concern is the estimate of total population size. This estimate is based on a density estimate from a single site, which is then extrapolated to represent all suitable habitat. The site where the density estimate was taken was not selected randomly, but was instead a site where multiple males were calling. This thus seems biased to be a higher density than average. Also, I don't think data from a single site should ever be assumed to represent all other sites. There needs to be replication.

Additional comments

The goal of this study was to identify suitable habitat for an endangered Brazilian frog, estimate densities of the frog, and from this reassess its conservation status. Prior to the study, the species was only known from a single site, and based on this work it has now been documented at seven additional sites. This is valuable information. Based on the eight known localities and habitat associations, the authors then estimate a total area of occupancy for the species. This is also valuable information, although I found the text somewhat confusing on this point (Lines 135-140). Even though this is clearly a calculation of area of occupancy, the authors are considering it extent of occurrence? Also, was the species only found in Montane Dense Ombrophilous Forest? If yes, then shouldn’t area of occupancy be limited to only this habitat type? I had trouble following the text, so I’m not sure of the answer to either of these questions.

The authors then move on to estimating population density. I found this estimate to be highly questionable to the point that I do not think it should be published. If I understand the procedure correctly, the authors found a site where multiple frogs were calling, counted the number of frogs within this area, and then extrapolated that density to the entire area of suitable habitat. Given that this site was selected because it had multiple males calling within it rather than being a randomly chosen area, it seems biased towards having a higher density than an average area of the same size. Also, this density estimate was only conducted at a single site with no replication. I object to the assumption that any individual site is representative of the entire species range. Any individual site could have a much lower or much higher density than the range-wide average. Certainly, the total population size estimated using this method seems farfetched. Do the authors honestly believe that there are 73 million individuals of the locally endemic frog?

I also do not understand how the authors concluded that the appropriate listing status is Vulnerable. They estimated an area of occupancy of ~148 km2. This would put the species in the Endangered category since area of occupancy is less than 500 km2. On the other hand, I do question whether the area of occupancy was estimated accurately. The authors surveyed 22 sites and found the frog at 7 of the sites. The survey protocol followed at each site (Lines 94-95) was impossible to follow. Thus, I have no idea how intensely each site was surveyed. If a frog was found, then clearly they are present, but if they were not found, how accurately does that represent absence? It is very difficult to prove absence, but some estimate of confidence would be very helpful. For instance, at the 7 sites where the frogs were found, were they found immediately? Were the remaining 15 sites searched for a much longer period of time and still no frogs were found? Or does the fact that the frogs use advertisement calls make it very simple to determine if they are present or not? Also, since the 22 survey sites were not picked with this species in mind (original search was for land mollusks), how likely is it that additional suitable sites outside of the current extent of occurrence were simply not searched? Looking at Figure 3, the current extent of occurrence seems to cut through a couple of the largest habitat patches, especially on the north and east sides, which I assume is where more of the mountains are located. It seems that there should at least be a discussion of how much suitable habitat likely remains unsearched, in which case maybe the species truly is Vulnerable rather than Endangered.

Line 25: Replace “In the anurans group” with “One group of anurans”.

Line 29: Delete “it”.

Line 30: Delete “threatened of extinction”.

Line 33: Change to “evaluated in loco impacts and potential impacts on the species’ population”

Line 34: Change “his” to “this”.

Lines 37-38: Change to “occurrence as 14,843.6 ha, density as one calling male per 4.05 m2, and a total”

Line 49: Change to “Anura is a priority group”

Line 59: Change “accessed” to “assessed”

Line 63: Aren’t the majority of species restricted to climatic envelopes? In other words, aren’t there some climate conditions outside of which they cannot persist? If so, then I think you must mean something more specific here about the tolerances of B. tridactylus.

Line 66: change “from” to “by”.

Line 70: Genus should be in italics.

Line 72: Change “which are associated with” to “exacerbated by”.

Line 78: Change “from” to “to”.

Line 79: Change “altitude” to “altitudinal range”.

Line 82: Change “was evidence that the species was threatened with extinction” to “led to the species being classified”.

Line 86: Change “over” to “on”.

Lines 94-95: I do not understand the sampling regime. Please explain this better.

Line 103: Not clear what you mean by “observed impacts” here.

Line 107: Insert “meters” after “five”.

Line 127: Change “amplitude” to “range”.

Lines 135-140: It is very important to differentiate between area of occupancy and extent of occurrence, as IUCN uses different thresholds for these two different quantities, but I don’t really follow how that was done.

Line 146: Delete “or, better, of one individual per m2.”

Lines 147-148: Delete “of two individuals per m2”

Line 169: Insert “an” before “advertisement”.

Line 170: Change to “we did not record the species at 15”.

Lines 172-173: If all localities are in this forest type, then why not limit the area of occupancy to only include areas with this habitat type that fall within the minimum convex polygon?

Line 173: Replace “connected through the” with “enclosed in a”.

Line 174: Change “resulting in” to “with”.

Line 176: Delete “previously”.

Line 183: The method by which total population size was calculated seems highly questionable. If I understand the method correctly, it sounds like an area with lots of males calling was first identified, and then the density of males within this area was determined. The density within this area was then extrapolated to be the same across the entire area of occupancy. However, if the area to sample was chosen because it had many calling males within it, it was presumably biased toward having a higher density than a randomly chosen region with the same area. If this method is to be used, it seems that random areas need to chosen first, and then numbers of frogs need to be counted within these random areas. Also, it seems critical that multiple sites be surveyed in this way rather than a single site. Any single site could have a non-representative density (either high or low) compared to other areas across the range.

Lines 190 and 192: Delete “limits”.

Line 197-198: How are fires going to change pasture to forest? I assume you mean the other way around.

Line 201: Replace “inland” with “interior”.

Line 204: What do you mean by “mischaracterizing”?

Line 213: It seems that this entire section should be in the Discussion rather than the Results.

Line 216: Change to “firefighting teams would serve to prevent”.

Lines 221-222: The goal of radio tracking mules is to stop palmiteiros? If yes, then this needs to be more clearly stated.

Lines 227-228: I don’t follow the logic here of brining the state closer to the people. How is this being achieved and why is it desirable?

Lines 265-267: This suggests that the minimum distance between Brachycephalus species is 4 km. Are they never sympatric?

Line 268: What do you mean by a sampling hiatus?

Line 271: Change “another 15 target points were unsuccessful” to “they were determined to be absent from another 15 target points”.

Line 272: How easily was the species detected when it was found? Were many individuals always found, or was the first individual always detected right away? In other words, how confident should we be that they were truly absent from the 15 sites where they were undetected?

Line 282: Change “sympathy” to “sympatry”.

Line 289: Change “superior to” to “higher than”.

Line 313: Insert “also” before “requires”.

Line 356: Change “IUNC” to “IUCN”.

Line 361: What does “conduct adjustment terms” mean?

Line 369: Change to “continues to suffer reduction and loss of habitat quality”.

Line 376: Why does it matter if other protected species are larger in body size?

Lines 377-379: Trying to introduce a new category of protected species in the last sentence of the paper seems strange. I don’t understand what would qualify a species as a fountain species.

Figure 3 caption: Change “state’s” to “state”.

Table 1: For Torre Embratel, the altitude range where animals were found does not fall completely within the altitude range that was searched.

·

Basic reporting

The English language should be improved throughout the manuscript to ensure that an international audience can clearly understand your text. Some examples where the language could be improved include Lines 138, 268-270, 282, 298, 308 – the current phrasing makes comprehension difficult. I suggest you have a colleague who is proficient in English and familiar with the subject matter review your manuscript, or contact a professional editing service. Latinized words (e.g., in loco) must be italicized (e.g, Lines 33 and 245).

The article includes sufficient introduction and background to demonstrate how the work fits into the broader field of knowledge. Relevant prior literature is appropriately referenced.

The structure of the article conforms to an acceptable format of ‘standard sections’. Figures are relevant to the content of the article, of sufficient resolution, and appropriately described and labeled (except see Additional Comments for issue with Figure 3 caption).

The study does not follow an ecological hypothesis, but rather is a review of the conservation status of a threatened amphibian species.

Experimental design

The knowledge gap being investigated is clearly identified, and statements are made as to how the study contributes to filling that gap. The research appears to have been conducted in conformity with the prevailing ethical standards in the field.

The greatest area for improvement lies with the methods used to calculate population size and extent of occurrence (EOO), with the later having implications for the proposed IUCN category and criteria. See Additional Comments for greater detail on these issues.

Validity of the findings

All underlying data have been provided. There is very thorough discussion as to concrete conservation actions needed for the species and its habitat. Conclusions are well stated and succinct, and link directly to the objective of the study which is to reevaluate the conservation status of Brachycephalus tridactylus and propose conservation measures.

Additional comments

Lines 173–180 (Geographic and Altitudinal Distribution): It is not clear how the extent of occurrence (EOO) was calculated and whether it was done so according to IUCN guidelines. Based on the text, it seems that the authors: 1) drew a minimum convex polygon around the known localities and calculated the area of this polygon as 53,289.0 ha (532.89 km2); 2) contracted the minimum convex polygon to exclude altitudes outside of the species’ altitudinal range, as well as non-forested areas; and 3) reported the updated EOO based on the revised range as 14,843.6 ha (148.436 km2). This second EOO value seems dubious given that step 2 essentially represents a range map for the species, and the area of a minimum convex polygon around this range map (IUCN’s definition of EOO) is much closer to the value provided in step 1 (ca. 530 km2) than the final EOO value provided by the authors (148.436 km2). Either EOO value would make the species eligible for the EN category under criteria B1ab (see Conservation Status section below for further detail on this), but I still find the author’s method for deriving their final EOO to be a bit puzzling and requires clarification. Upon examination of the caption for Figure 3, I suspect that the authors may be using ‘EOO’ interchangeably with ‘species geographic range’. The light grey polygons in this figure represent the species’ presumed range, not the EOO. The EOO, according to IUCN definition, is a measure of the spread of risk calculated by a minimum convex polygon around all sites of occurrence – in this case, the solid white line is the EOO.

Lines 182–186 (Density and Population Size): Density was measured at a single site based on calling activity for 3 hours. I think that extrapolating these results to the entire range to obtain a total population size has a lot of limitations that are not discussed, such as the fact that the species is unlikely to be evenly distributed throughout the range and therefore their population size estimate may represent an overestimate.

Lines 255–260 (Conservation Status): The proposed IUCN Category and Criteria of VU B1ab(i,ii,iii) does not agree with the data provided in this manuscript. First, the EOO value is below the threshold for EN under Criterion B1. Second, he authors state that the geographic range is severely fragmented, but it is not clear whether the population is severely fragmented according to the IUCN definition (ie. “most (>50%) of its individuals are found in small and relatively isolated subpopulations”). If the population does meet the IUCN definition of a severely fragmented population, which is one of the requirements of using Criterion B, then the text should specifically state this information rather than leaving it up to the reader to speculate based on a seemingly fragmented distribution. Considering this species has an EOO of <5,000 km2, a severely fragmented population, and ongoing decline in the extent of occurrence (i), area of occupancy (ii), and area, extent and/o quality of habitat (iii), I would argue that EN B1ab(i,ii,iii) is a more appropriate listing for this species than VU.

Lines 304–306 (Conservation): I think it is important to clarify that B. tridactylus has not yet been assessed by IUCN and that the “previous classification of the species as Vulnerable” was an assessment by Bornschein, Pie & Teixeira (2019) not the IUCN Red List.

Reviewer 3 ·

Basic reporting

This manuscript has original and fundamental data for the evaluation and protection of a species of anuran endemic to Brazil.
The manuscript is very well written, has a review of clear terms, and has an excellent writing structure.
Bibliographic references are well up-to-date, except for those highlighted in the correction file.
All data, tables, and figures are well produced and contain important information for the complete understanding of the manuscript. I made some suggestions for improvements.

Experimental design

The manuscript presents an appropriate field methodology and data collection and analysis.
The proposal is well constructed and is essential in the level of detail presented to support a species assessment.
The authors demonstrated a great effort to obtain and analyze the data. Reflecting on important results and discussions.

Validity of the findings

This manuscript can serve as a citation for many other manuscripts dealing with species with some degree of threat.
I emphasize that the degree of detail of the information presented, with the analyzes and conclusions presented, make this manuscript an important paper for this renowned journal, in addition to serving as a model for future papers with the same focus.

Additional comments

With the details in the previous fields, I indicate that the paper should receive a major revision and few minor revisions to adapt some terms, understanding of paragraphs, and the conclusion.
Despite few and localized corrections, this manuscript must be passed on to the authors for the adequacy of this information.
This article deserves to be revised after corrections, since it brings new and important information for the protection of a species that is threatened and that, without proper subsidy, may be extinct in a short time.

Annotated reviews are not available for download in order to protect the identity of reviewers who chose to remain anonymous.

---

## Round 0.2 · Major Revisions

Your paper has improved significantly since the first version. However, one of the referees highlighted several points for improvement in your revised article. I agree with all of them.

Please, review your paper again and follow all the reviewer's suggestions. Focus on the information that you have and avoid generalizations that are not technically sound.

I look forward to receiving a new version of your manuscript.

Reviewer 1 ·

Basic reporting

This is fine.

Experimental design

There was still a lack of detail on how surveys were conducted at each individual survey site.

Validity of the findings

Despite consensus among the reviewers that improvements needed to be made in the estimate of total population size (or rather that it should be dropped) and that the extent of occurrence estimate was being mixed up with area of occupancy, the authors did not alter their original approach.

Additional comments

I was very disappointed with this revision. It seemed that the reviewers gave very clear, largely consistent suggestions, most of which were ignored. There were two main improvements. First, the authors changed the recommended status from Vulnerable to Endangered. This was good, since all three reviewers made this recommendation. Second, the distribution of the survey sites was explained much better using Figure 1. I am now confident that the authors searched at least the majority of the potential suitable habitat.

Unfortunately, many other recommendations were not followed. First, the authors persisted in providing an inaccurate estimate of total population size. Two of the reviewers pointed out that this estimate was problematic. I appreciate that the authors went to the trouble of surveying a second site. I also understand the logistical limitations to conducting surveys at completely randomized sites. However, it seems the solution should be to forego making any population estimate at all rather than making an inaccurate one. The fact that the authors found such a different density at their second survey site compared to their first one, which resulted in a greater than 10-fold change in their estimate of total population size, should have indicated to them that they are likely still far off from the true number. The authors themselves state that this species has a patchy distribution, but in their estimate calculate that the same density applies across the entire area of occupancy. Do these two statements/approaches not strike them as contradictory?

Second, while the authors have now clarified the methods used to select the survey sites, there is now even less information on how the survey was conducted at each individual site. I previously encouraged the authors to provide more details about their field methods, especially an indication of how quickly they found the species at sites where it was recorded relative to the total amount of time spent searching each site. This would provide some indication of how likely it is that the species is truly absent from sites where it was not found.

Third, the authors persisted with their previous calculation of extent of occurrence, even though all three reviewers found this to be problematic. The authors’ approach seems to be to first calculate a minimum convex polygon around all of the known localities, which is equivalent to the extent of occurrence. They then remove areas within this polygon that have unsuitable elevations or habitat types. This would lead to a calculation of area of occupancy. The authors then state that they think the area of occupancy should be even smaller, so they call their estimate of area of occupancy the extent of occurrence. If the authors aren’t confident in their estimate of area of occupancy, then they should stick with their original estimate of extent of occurrence using the minimum convex polygon.

Line 39: What does “67.0.2” mean?

Lines 41-42: Change to “We recommend changing the species’ conservation status from Vulnerable to Endangered”. Only IUCN can actual change the species’ status, so this is just a recommendation.

Line 64: Insert “usually” before “heard”.

Lines 65-66: My point from before is that you could say this about most species in the world. I think you must really want to say something more specific than this, such as the kind of climate envelope the group is restricted to.

Lines 74-75: If the small distributions are natural, then they don’t represent losses in areas of occupation or habitat quality. Maybe you mean that the small distributions in conjunction with losses in areas of occupation and quality of habitat are the main cause of threat to the species?

Line 95: Change “from” to “around”.

Line 96: Change “specialization” to “specificity”.

Line 100: Change “parts” to “quadrants”.

Line 101: Change “mountain” to “mountainous”.

Lines 113-114: Since everything before this has been described as an intentional search for B. tridactylus, this doesn’t make any sense.

Line 115: Delete “local anesthetic”.

Line 119: Change to “from its type locality”.

Lines 135-137: Exactly. This is why extrapolating density from a site where you found them to the entire range is a bad idea. You’ve just said here that there are some areas where density is zero. However, in your calculation, you are saying that all areas have the same density as your survey areas.

Lines 137-139: I am sympathetic to the reality that randomizing survey locations would not have been possible. If this means that you can’t get an accurate estimate of total population size, then don’t do one at all. Stick to providing information about geographic extent and threats. That’s the useful information anyway. An unreliable estimate of total population size isn’t helping anyone.

Lines 144-152: This reads as Intro or Discussion material. If this relates to data that you yourselves collected while in the field, then you need to be clearer about how that was done.

Lines 165-176: This seems like flawed logic. You followed the procedure for calculating area of occupancy, but then because you think it is a bad estimate for area of occupancy you call it extent of occurrence. I actually think you do have a good estimate of area of occupancy. However, if you don’t agree, then just stick to calculating extent of occurrence using the minimum convex polygon. Don’t try to refine that estimate by excluding inappropriate habitats within the polygon, since that procedure is part of calculating area of occupancy, not extent of occurrence.

Line 186: So you searched areas between 700-1140 m and found the species across almost that entire elevational range? If so, saying that the species is restricted to this elevational range seems flawed. If you didn’t search outside of this range, then you don’t have any information on whether the species occurs there or not. You clearly state that you didn’t search any sites above 1140 m, so you definitely don’t have information about whether this is the upper elevational limit for the species? How many sites between 700-714 m did you search and not find the species? It would need to be a large fraction of your total absence sites (~8 out of the 15) to be confident that this was the true lower elevational limit.

Line 204: The fact that the researchers found such different densities at the two sites they surveyed indicates that the distribution of the species is indeed patchy, especially since the lower density was found at the larger site. Doesn’t this suggest to the researchers that the larger they made the area that they surveyed, the lower a density they would find? Thus, estimating a mean density from this data and applying it across the entire range seems flawed. I’m sure that the new population estimate is much closer than the previous one, but I would think the fact that the estimate has changed more than 10-fold by the addition of just one more survey site would suggest to the researchers that the new estimate is still likely to be way off the true number.

Line 233: As I said previously, this whole section belongs in the Discussion. These are not things that you observed in the field. These are recommendations based on what you observed in the field.

Lines 247-249: Won’t the people likely be resentful of the punitive acts and thus less likely to cooperate with conservation of the state park? It seems like there are two conflicting approaches here. One could either educate the people about the importance of the environment and try to enroll them as collaborators in protection. Or, one could punish them for violations and force them to follow the conservation laws. Aren’t these two separate approaches rather than the same one?

Line 282: To meet these classifications you also need to show that extent of occurrence is below a specific cutoff. You should report the extent of occurrence that you calculated and compare it to this number.

Lines 287-289: Do you mean something more specific here, such as gaps between the distributions of sister species of Brachycephalus? Below you describe sympatry in the genus, so clearly gaps between the geographic distributions of two Brachycephalus species can actually be zero.

Line 292: Change “from” to “around”.

Lines 293-295: This is not clear.

Line 357: This is unclear.

Line 372: This is unclear.

Lines 379-380: Now reads like B. tridactylus occurs in the entire Atlantic Rainforest. Need to clarify what you really mean.

Line 391: This is unclear.

Lines 407-408: Why are these effects going to go beyond the extent of occurrence of the species? Are they going to be implemented in other areas where the species isn’t even present? Wouldn’t they benefit other species that co-occur with the focal species?

Lines 408-410: I still think it is strange to introduce this new concept in the final sentence of the paper. There is no indication of how this approach would actually work.

·

Basic reporting

no comment

Experimental design

no comment

Validity of the findings

no comment

Additional comments

After reviewing this new version, I appreciate the effort made to address my comments, I am pleased with the responses of the authors to some of the observations I made. Having said this, I have no comments for this new version of your draft.

---

## Round 0.3 · accepted · Accept

I believe your addressed most of the main comments by the referees.